# Concordance Analysis between the Segments and the Overall Performance in Olympic Triathlon in Elite Triathletes

**DOI:** 10.3390/biology11060902

**Published:** 2022-06-11

**Authors:** Javier Olaya-Cuartero, José Fernández-Sáez, Ove Østerlie, Alberto Ferriz-Valero

**Affiliations:** 1Faculty of Health Sciences, Isabel I University, 09003 Burgos, Spain; javier.olaya@ui1.es; 2Terres de l’Ebre Research Support Unit, Jordi Gol i Gurina University Institute for Primary Health Care Research (IDIAPJGol), 43500 Tortosa, Spain; jfernandez@idiapjgol.info; 3Research Unit, Terres de l’Ebre Territorial Management, Catalan Institute of Health, 43500 Tortosa, Spain; 4Facultat de Enfermería, Campus Terres de l’Ebre, Universitat Rovira i Virgili, 43500 Tortosa, Spain; 5Department of Teacher Education, Faculty of Social and Educational Science, NTNU–Norwegian University of Science and Technology, 7491 Trondheim, Norway; 6Department of General and Specific Didactics, University of Alicante, 03690 Alicante, Spain; alberto.ferriz@ua.es

**Keywords:** sports performance, triathlon disciplines, indicator of performance, endurance, competition

## Abstract

**Simple Summary:**

Due to the complexity of the triathlon, it is difficult to analyse overall performance. To date, the analysis of performance in triathlon has been widely studied through time or position in the three segments and in the overall result, which is what defines the medals and the goal of the competition, but it can have some limitations. As an alternative, the purpose of this study is to analyse the concordance between each of the triathlon segments (swimming, cycling, and running) and the overall performance in the Olympic triathlon in elite triathletes. The main results of the present study show that performance in the cycling segment presents the best concordance with overall performance. In conclusion, the cycling performance indicator could be an alternative to anticipate the overall performance in the competition. For this reason, the cycling segment would not be a smooth transition toward running in the Olympic distance event.

**Abstract:**

To date, the performance in triathlon has been measured through time or position. Although this is what defines the medals and the goal of the competition, it can have some limitations. As an alternative, the purpose of this study is to assess the degree of concordance of performance between each of the triathlon disciplines with overall performance through the triathlon performance indicator for the Olympic distance event. The official results from the World Triathlon Series for Olympic distance events from 2000 to 2019 were examined. A total of 11,263 entries were analysed, 6273 corresponding to elite men and 4990 to elite women. Moderate agreement was found between the running performance and overall performance in both elite men ICCa = 0.538 and elite women ICCa = 0.581. Moreover, moderate agreement was found between swimming performance and overall performance in both elite men ICCa = 0.640 and elite women ICCa = 0.613. Finally, good agreement was found between cycling performance and overall performance also in both elite men ICCa = 0.777 and elite women ICCa = 0.816. The main results of the present study show that the cycling performance indicator could be an alternative to anticipate the overall performance in the competition for the Olympic distance event.

## 1. Introduction

The triathlon is a sport that has grown very rapidly over the past two decades, particularly in short events [1], increasing the attention of coaches and researchers in seeking to obtain medals in the Olympic Games and win economic awards in the World Triathlon Series (WTS). The opening sport at the Olympic Games in Sydney 2000 was the triathlon due to the gained popularity and official recognition of multi-sport activities [2]. The determination of triathlon performance factors has been widely studied for Olympic distance events [3]. However, due to the complexity of the triathlon, it is difficult to analyse overall performance [4,5], even more so in elite athletes, because the sports performance is particularly complex and multifactorial [6].

Theoretically, performance is a concept that refers to the relationship between the means used to achieve something and the overall result. Practically, any coach needs to have a useful tool to measure the performance of their athletes to use the most suitable tactics in competition. To be able to apply adequate measuring instruments, it is necessary to carry out a process of translation of theoretical concepts into empirical language—that is, to replace what cannot be observed with that which is accessible to observation and, therefore, to measurement instruments [7]. This process of transition from the theoretical to the empirical is known as the process of instrumentalisation, where the information provided by several indicators is synthesised into a complex index using a mathematical function. To define a complex index—and, in particular, a performance index—important conceptual, analytical, and empirical decisions must be made [8]. Therefore, the creation of these indexes generates a continuous and in-depth methodological debate, which has resulted in the development of many approaches in recent years [8,9]. The difference between all these indices lies, fundamentally, in the choice of dimensions or variables and in the way they are related, in such a way that, when it comes to measuring, answers can be given to the questions of what, how, when, and where [10].

To date, to analyse triathlon performance, the position of the segments and the influence on the overall result in the Olympic distance event has been studied [11]. However, performance is not always well represented by the triathlete’s position [12] because the position presents some limitations when measuring performance. The influence of each segment time on the overall result of the competition also has been the subject of study in numerous investigations. Piacentini et al. [13] concluded that the overall race time for the Olympic distance event varies between 106 and 100 min for elite males and between 119 and 121 min for elite female athletes. However, the comparison between triathlon competitions is difficult because, even considering a race performed on the same course, there may be many differences depending on the current, elevation profile, and climate, among other aspects [14]. Moreover, analysing together competitions over periods of 26 years, including competitions where drafting was allowed and competitions where drafting was forbidden [15], would be another challenge in interpreting the performance through time. Nonetheless, numerous studies have used time to analyse the influence of the segments in the Olympic distance triathlon, reaching similar conclusions, pointing out that running performance is the primary determinant of success in high-level Olympic distance triathlon races [1]. Figueiredo et al. [15] found the highest contribution to overall performance in the running segment, suggesting that this segment should be the focus in the preparation for the Olympic distance event. Moreover, the running segment had the strongest relationship with finishing time in the Olympic distance event [16]. In addition, in the running segment, it was found that there was the greatest variation in times for all-male world championship triathletes [17]. Some of these investigations also show that bike leg exercise seems to be a smooth transition toward running [13]. Concerning the time of the swimming section, it has been found to show the lowest correlations with overall finishing time [16]. Similarly, in Olympic distance events, as in Ironman distances, swimming’s contribution was lower compared to other disciplines for both sexes [15,18].

Another point to keep in mind is the difference between the sexes in the triathlon. It has been explained by differences in physiological and anthropometric characteristics [19], which also has been associated with performance variables in young triathletes [20]. Regarding time, future studies are required to clarify why the sex difference in running is greater compared to swimming and cycling [21]. However, it should be noted that the times of both sexes could never be compared to interpret the performance of elite men and elite women because men and women never compete together.

As an alternative to the analysis of performance through time or position, which shows several limitations, different solutions have emerged, such as the relative performance index in the triathlon [22]. This index has been used in triathletes with highly specific characteristics, i.e., they have been top 10 in at least five Ironman competitions. Therefore, to the best of our knowledge, it would be difficult to perform an external validation of this index. Methodologically, it is similar to the concept of effect size in that all triathletes start from a common state, which is the average time of all 24 triathletes, and the index measures the effect of their competition concerning the average of all. In other words, it measures by how many standard deviations of all triathletes the time of each triathlete differs from the average of all triathletes.

Hence, it would not be appropriate to study agreement using Pearson’s linear correlation coefficient, since, as its name indicates, it measures linear correlations. This means that it would not be an adequate measure of the degree of agreement between two measurements, since, if two instruments measure systematically different quantities, the correlation may be perfect (r = 1), even though the agreement is null. However, correlation coefficient analyses are mostly used to determine which segment of the triathlon is most associated with the overall performance in the competition, but these would not be the most appropriate since calculation depends on variability between subjects [23] and is sensitive to the range of values under study [24], i.e., larger ranges of measurement values will result in overestimated coefficients [25]. Thus, a high value of the correlation coefficient does not necessarily indicate a good degree of agreement between measurements, since this coefficient does not detect systematic or random differences between measures. Therefore, concordance studies would be the most appropriate to determine the degree of agreement between a measure and the standard measure, so that a measure can be used instead of the standard measure.

In this case, the purpose of the study was to assess the degree of concordance of performance between each of the triathlon disciplines (swimming, cycling, and running) with overall performance in Olympic distance events in elite triathletes. For this purpose, the authors have proposed, as a gold standard, a performance indicator for triathlon that has previously been used to study the contribution of segments to overall results in elite triathletes in sprint distance [5]. Moreover, this performance indicator has been used to analyse the differences in young triathletes regarding their performance within each competitive group [4].

## 2. Materials and Methods

### 2.1. Participants

All data were the official results from WTS events for elite men and elite women at Olympic distances from 2000 to 2019. A total of 11263 entries were analysed, 6273 corresponding to elite men and 4990 to elite women, excluding all competitions where there was no information about segments and transition times or the Olympic distance was altered due to technical or environmental issues.

### 2.2. Procedures

A concordance study was conducted, measuring the continuous variables through absolute agreement and consistency. The time of the swimming, cycling, and running segments, time in transitions, and the overall time of the competitions were collected as data for the calculation of the performance indicator for all segments, transitions, and the overall performance indicator. The research design was based on an observational model without interference with the natural context, including all segment times of the WTS races, recorded through a chip-based timing system to obtain a highly accurate value for individual performance [26]. Therefore, it would be possible to use the performance indicator in the triathlon as a dependent variable to analyse performance in elite male and elite female triathletes. This variable is expressed from 0 to 10,000, where 10,000 is the best segment time and thus the best performance:$$\mathrm{OPI}=\frac{\mathrm{Winner}\;\mathrm{time}}{\mathrm{Personal}\;\mathrm{time}}\;\times \;10,000$$

A performance indicator is provided through this calculation for each segment and transition—swimming performance indicator (SPI), cycling performance indicator (CPI), and running performance indicator (RPI)—and for each swim-to-cycle transition (T1PI) and cycle, cycle-to-run transition (T2PI), and for the overall performance indicator (OPI). This performance indicator has previously been used to study the contribution of segments to overall results in elite triathletes in sprint distance events [5] and to analyse the differences in young triathletes regarding their performance within each competitive group [4].

### 2.3. Performance Indicator: Internal Validation

To study whether the performance indicators were distributed according to a normal distribution, the Kolmogorov–Smirnov normality test was applied. None of the performance indicators followed a normal distribution (*p* < 0.001 for all performance indicators), justifying nonparametric analyses.

To perform the internal validation of the overall performance indicator and the performance indicators of the three different segments, three subsamples were taken, selected at random—first 10% (Table 1), then 25% (Table 2), and finally, 75% (Table 3) of the total sample—and the analyses were replicated, obtaining identical results.

### 2.4. Statistical Analysis

Intraclass correlation coefficients of absolute agreement (ICC_a_) and consistency (ICC_c_) were calculated for single measures, with the corresponding 95% confidence interval (IC95%), to analyse the concordance of the performance of each segment with the overall performance of the competition. Specifically, the absolute agreement intraclass correlation coefficient (ICC_a_) was calculated, which considers any difference between performances as a discordance, while the consistency intraclass correlation coefficient (ICC_c_) does not consider the constant differences between performances. The ICC_a_ and ICC_c_ take values between 0 and 1, corresponding to the maximum possible agreement as a value of ICC = 1. To interpret the magnitude of concordance between measurement variables, the following criteria were adopted: <0.50 (poor), 0.51–0.75 (moderate), 0.76–0.90 (good), >0.90 (excellent) [27]. To interpret the Spearman range correlation coefficient, the same criteria were adopted.

The Bland–Altman method allowed the comparison between the swimming, cycling, and running performance indicators and the overall performance indicator of the competition. This graphical representation method, proposed by Bland and Altman, was used to assess the degree of agreement between the performance indicator of the swimming, cycling, and running segments and the overall performance indicator, using the mean values of the performance indicators against their differences. The systematic error that quantifies the extent to which the performance of each segment overestimates or underestimates the overall performance is represented by the average of the differences in the values [28]. Moreover, it reflects the precision with which the performance indicator of each segment estimates the overall performance, representing the degree to which the values are grouped around the average, quantified through the interval of ± 1.96 standard deviations of the differences between the two measurements systems. The software Statistical Package for The Social Sciences (v.24.0 SPSS Inc., Chicago, IL, USA) and a Microsoft Excel spreadsheet were used to analyse data. Significance was accepted at *p* < 0.05.

## 3. Results

Table 4 summarises the relationship between the performance indicators of the segments and the overall performance indicator of the competition for elite men and elite women. Regarding performance, moderate and good agreement was found between the performance in the segments and the overall performance in both sexes. Sorting the segments from lowest to highest agreement of the segment with overall performance, similar results were shown in both sexes. Moderate agreement was found between the running performance and overall performance in both elite men ICC_a_ = 0.538, IC95% = (0.179–0.788) and elite women ICC_a_ = 0.581, IC95% = (0.188–0.820). Similarly, moderate agreement was found between swimming performance and overall performance in both elite men ICC_a_ = 0.640, IC95% = (0.622–0.658) and elite women ICC_a_ = 0.613, IC95% = (0.588–0.637). Good agreement was found between cycling performance and overall performance also in both elite men ICC_a_ = 0.777, IC95% = (0.700–0.828) and elite women ICC_a_ = 0.816, IC95% = (0.633–0.890). Therefore, performance in the cycling segment has the best concordance with overall performance. In elite men, poor, moderate, and good correlations were found between swimming (ρ = 0.424, *p* < 0.001), cycling (ρ = 0.535, *p* < 0.001), and running position (ρ = 0.843, *p* < 0.001) and overall performance in the competition, respectively. Similarly, in elite women, poor, moderate, and good correlations were found between swimming (ρ = 0.425, *p* < 0.001), cycling (ρ = 0.709, *p* < 0.001), and running performance (ρ = 0.789, *p* < 0.001) and overall performance in the competition, respectively. The performance in the transitions showed little concordance and a poor correlation with overall performance in both elite men and elite women.

Figure 1 shows the distribution of the overall performance indicator concerning the performance indicator of the swimming, cycling, and running segments according to sex. The cycling segment, where performance is most similar to overall performance in terms of performance values, is distributed along the diagonal, indicating that the two are most similar. There is more dispersion in the swimming segment in comparison to the diagonal. In the running segment, it can be seen that the performance distribution is above the diagonal, indicating that low performance values in this segment correspond to high values of the total performance. In addition, this is true for both men and women. These results are the same for both sexes.

Figure 2 and Figure 3 show the agreement between the performances of the different segments and the transitions with the overall performance of the competition using the Bland–Altman plots. Figure 2 shows that the swimming performance overestimates the overall performance in men by 0.09 but underestimates the overall performance in women by 44.12. The performance score of 9500 is passed when the best agreement is seen in both sexes. In the case of the cycling performance, in both sexes, this performance underestimates the overall performance (77.76 in men and 110.62 in women), having also narrower concordance intervals than in the rest of the sections. After the performance value of 9500 is reached, the concordance increases. It is the running performance that agrees less with the overall performance since it underestimates it by 468.14 and 477.20 in men and women, respectively. In the two transitions, both in men and women, there is little agreement between the performance of the latter and the overall performance (Figure 3).

## 4. Discussion

This study shows that there is a high degree of agreement between the cycling segment performance and the overall performance. However, it is also worth explaining the results obtained in the other segments. Firstly, regarding the swimming segment, moderate agreement was found between swimming performance and overall performance both in elite men and elite women. Therefore, this discipline does not show the lowest agreement with overall performance in the triathlon race; rather, running performance shows this. The moderate agreement concerning the performance explains that the swimming segment is important to ensure successful overall performance in elite triathletes. However, the poor correlation of swimming position with overall position might explain that this segment is not a determinant of staying in the first position in the competition. From the point of view of triathletes and coaches, it could be explained that the position in which the athlete exits the water is less important than their proximity to the first triathlete. This is partially supported by previous investigations [13], which show that although the contribution of the swimming segment might be low, the strategic positioning within this segment may be critical to overall race performance [13]. Similarly, Landers et al. [11] showed that 90% of male winners and 70% of female winners exited in the first group of swimmers. This could explain why the agreement in elite men ICC_a_ = 0.640 (0.622–0.658) is slightly higher than in elite women ICC_a_ = 0.613 (0.588–0.637).

Secondly, the highest ICC_a_ for overall performance is found in the cycling segment, both in elite men ICC_a_ = 0.777 (0.700–0.828) and elite women ICC_a_ = 0.816 (0.633–0.890). This means that good cycling performance would explain good overall performance. Therefore, the cycling performance indicator could be key to anticipating overall competition performance. For this reason, the cycling segment would not be a smooth transition toward running at Olympic distances, as Piacentini et al. [13] point out. As with the previous segment, the position in which the triathlete finishes the cycling segment is not as important as how close to first place the triathlete is when they finish and start running. However, to achieve a deeper analysis of the results, it is worth noting that the findings of the present study show that the position in the running segment is the most related to the overall position in both elite men (^d^ρ = 0.843) and elite women (^d^ρ = 0.789). In contrast, the performance in this segment is the least consistent with the overall performance in the competition in both elite men and elite women. This could be explained by the fact that the running segment is the last; therefore, the athlete’s proximity to first place is less important as their position in this segment will have a greater influence on their overall position in the competition. Therefore, by using the triathlon performance indicator used in this study, it is possible to differentiate between the performance and position of the segment and the overall position. These results are coherent considering that, in the Olympic distance event, drafting is allowed and the triathletes arrive in groups at the second transition; thus, the position for cycling is not a determinant of the overall position, but better performance in the cycling segment, where the triathlete joins the first group (and cuts time from the first group), is. Moreover, it is worth noting that the largest range in the IC95% is found in the running segment, both in elite men (0.179–0.788) and elite women (0.188–0.820). These results could be especially useful for coaches because a greater dispersion of the data could mean that there could be a greater range of improvement in this segment. This can also be explained by the wide range of the limits of the Bland–Altman charts in this segment (Figure 2). In the same way, to date, the results of the studies in which the time or the position have been used to interpret the contribution of performance in each segment to overall performance have supported these results. Horne [16] noted that the running segment had the strongest relationship with finishing time in standard racing in the age-group World Championship. Similarly, Vleck et al. [29] determined that running may be the strongest predictor of overall performance in Olympic distance events. According to the results of the present study, and considering that the triathlon is a complex sport, it is important to analyse each segment globally.

Thirdly, poor agreement and correlations are found between performance and position in both transitions and overall performance and overall position. This is supported by previous investigations in Olympic distance events, which found a low correlation between T1 and the overall result [30]. However, given the wide range of transition confidence intervals, this could be the part of the race with the most extensive range of improvement. The importance of achieving a good transition is more closely linked to tactical importance [31].

## 5. Conclusions

To conclude, it should be noted that there is concordance between the Olympic distance triathlon segments performance and overall performance in the competition. Specifically, the cycling performance indicator could be an alternative to anticipate the overall performance in the competition in the Olympic distance. Therefore, the cycling segment would not be a smooth transition toward running at Olympic distances.

## 6. Strengths, Limitations, and Future Lines of Research

The main practical application is that the triathlon performance indicator allows the coach to interpret the performance of the athlete concerning each segment and the overall result. Therefore, coaches can identify the strengths and weaknesses of their triathletes in each of the three segments using the triathlon performance indicator.

Moreover, several strengths are highlighted by this tool in comparison with other performance indicators listed above. From a practical point of view, the triathlon performance indicator proposed in this study does not require large investments of time or money (i.e., it could be provided in an Excel file template). Another important strength to note is that it would also be possible to differentiate the performance between two triathletes (i.e., 1st and 2nd) in triathlons of all distances, from super sprint to Ironman distance, and it is applicable for both males and females.

External validation of the performance indicator is another major strength of the study; these results support the external validation of the performance indicator at Olympic distances but also at sprint distances [5]. Specifically, in the present study, external validation is carried out because the sample is composed entirely of elite triathletes who have competed in world-ranking races from 2000 to 2019 (without considering the races excluded from the study), which means that these data can be extrapolated to elite triathletes.

The main limitations of this study are related to the use of the performance indicator. This variable is limited to winner time because it is determined by this individual. However, in elite competitions, as in this study, high performance is always expected from competition winners. In this way, using this indicator to analyse the triathlete’s performance in the competition would still be more suitable than using the average time of all triathletes, as proposed in previous research [32].

Future studies will be required to validate this index both internally and externally with other types of participants, such as popular triathletes, popular and elite athletes, trail runners, etc.

## Figures and Tables

**Figure 1 biology-11-00902-f001:**
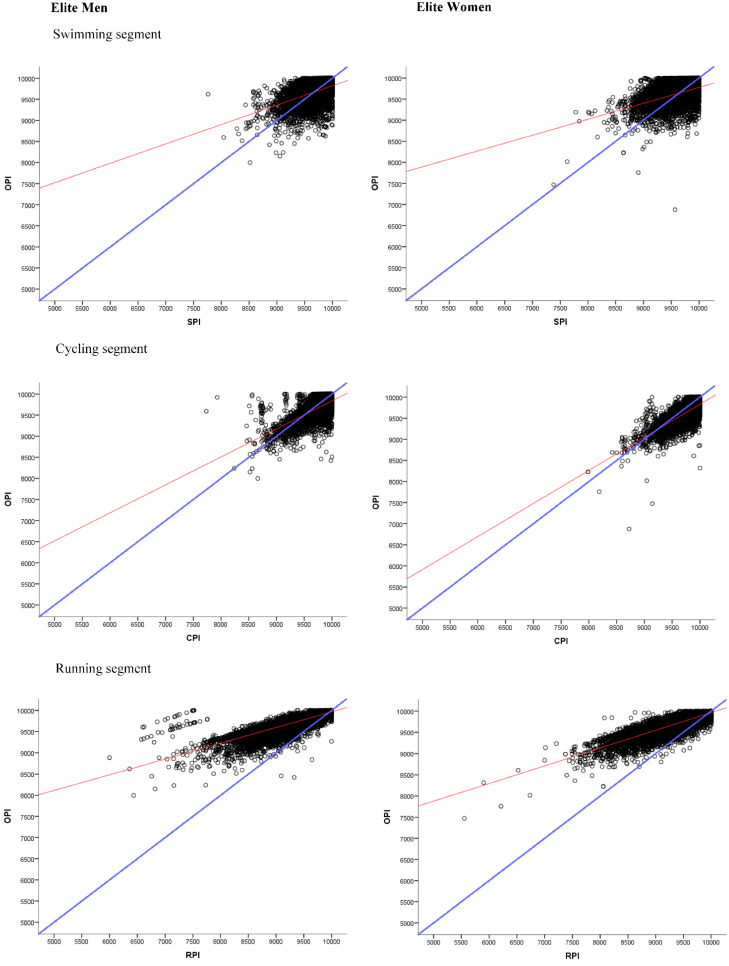
Distribution of the overall performance indicator (OPI) concerning the swimming performance indicator (SPI), cycling performance indicator (CPI), and running performance indicator (RPI).

**Figure 2 biology-11-00902-f002:**
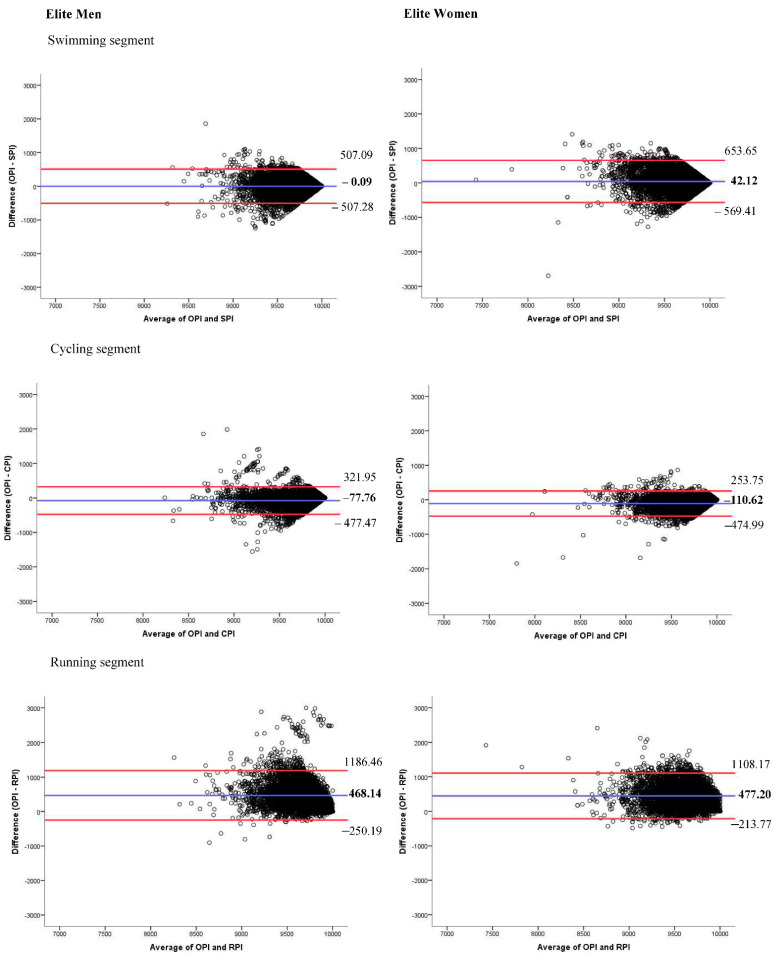
Bland–Altman plots between swimming performance indicator (SPI), cycling performance indicator (CPI), and running performance indicator (RPI) and overall performance indicator (OPI). The red lines represent the upper and lower 95% limits of agreement, whereas the blue line represents the bias.

**Figure 3 biology-11-00902-f003:**
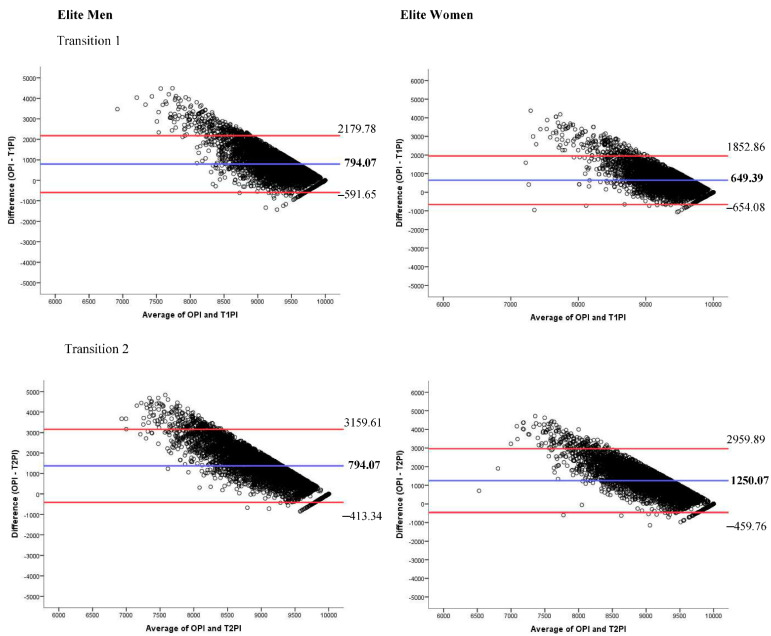
Bland–Altman plots between transition 1 performance indicator (T1PI) and transition 2 performance indicator (T2PI) and overall performance indicator (OPI). The red lines represent the upper and lower 95% limits of agreement, whereas the blue line represents the bias.

**Table 1 biology-11-00902-t001:** Random selection of the subsample representing 10% of the total sample.

	^a^ ICC_a_ (IC95%) ^b^	*p*	^c^ ICC_c_ (IC95%)	*p*	^d^ ρ	*p*
**Elite Men**						
SPI	0.630 (0.567–0.683)	<0.001	0.629 (0.566–0.683)	<0.001	0.436	<0.001
T1PI	0.147 (0.078–0.328)	<0.001	0.278 (0.154–0.384)	<0.001	0.261	<0.001
CPI	0.783 (0.683–0.845)	<0.001	0.809 (0.777–0.837)	<0.001	0.547	<0.001
T2PI	0.082 (0.072–0.223)	0.002	0.213 (0.074–0.330)	0.002	0.248	<0.001
RPI	0.564 (0.206–0.815)	<0.001	0.793 (0.758–0.823)	<0.001	0.859	<0.001
**Elite Women**						
SPI	0.639 (0.569–0.99)	<0.001	0.644 (0.574–0.702)	<0.001	0.449	<0.001
T1PI	0.170 (0.061–0.349)	<0.001	0.280 (0.138–0.399)	<0.001	0.219	<0.001
CPI	0.823 (0.619–0.901)	<0.001	0.866 (0.840–0.888)	<0.001	0.710	<0.001
T2PI	0.123 (0.101–0.318)	<0.001	0.313 (0.175–0.428)	<0.001	0.321	<0.001
RPI	0.584 (0.196–0.825)	<0.001	0.799 (0.760–0.832)	<0.001	0.791	<0.001

SPI, Swimming Performance Indicator; T1PI, Transition 1 Performance Indicator; CPI, Cycling Performance Indicator; T2PI, Transition 2 Performance Indicator; RPI, Running Performance Indicator; OPI, Overall Performance Indicator; ^a^ Absolute Agreement Intraclass Correlation Coefficient; ^b^ 95% Confidence Intervals; ^c^ Concordance Intraclass Correlation Coefficient; ^d^ Spearman Range Correlation Coefficient.

**Table 2 biology-11-00902-t002:** Random selection of the subsample representing 25% of the total sample.

	^a^ ICC_a_ (IC95%) ^b^	*p*	^c^ ICC_c_ (IC95%)	*p*	^d^ ρ	*p*
**Elite Men**						
SPI	0.645 (0.608–0.679)	<0.001	0.645 (0.608–0.679)	<0.001	0.442	<0.001
T1PI	0.081 (0.047–0.195)	<0.001	0.163 (0.074–0.243)	<0.001	0.159	<0.001
CPI	0.769 (0.685–0.824)	<0.001	0.793 (0.771–0.812)	<0.001	0.557	<0.001
T2PI	0.080 (0.070–0.218)	<0.001	0.226 (0.142–0.301)	<0.001	0.273	<0.001
RPI	0.534 (0.185–0.787)	<0.001	0.754 (0.728–0.777)	<0.001	0.834	<0.001
**Elite Women**						
SPI	0.583 (0.532–0.628)	<0.001	0.587 (0.538–0.631)	<0.001	0.426	<0.001
T1PI	0.171 (0.050–0.341)	<0.001	0.282 (0.195–0.360)	<0.001	0.243	<0.001
CPI	0.803 (0.587–0.887)	<0.001	0.850 (0.832–0.866)	<0.001	0.713	<0.001
T2PI	0.121 (0.096–0.303)	<0.001	0.295 (0.209–0.372)	<0.001	0.333	<0.001
RPI	0.576 (0.184–0.816)	<0.001	0.786 (0.761–0.809)	<0.001	0.782	<0.001

SPI, Swimming Performance Indicator; T1PI, Transition 1 Performance Indicator; CPI, Cycling Performance Indicator; T2PI, Transition 2 Performance Indicator; RPI, Running Performance Indicator; OPI, Overall Performance Indicator; ^a^ Absolute Agreement Intraclass Correlation Coefficient; ^b^ 95% Confidence Intervals; ^c^ Concordance Intraclass Correlation Coefficient; ^d^ Spearman Range Correlation Coefficient.

**Table 3 biology-11-00902-t003:** Random selection of the subsample representing 75% of the total sample.

	^a^ ICC_a_ (IC95%) ^b^	*p*	^c^ ICC_c_ (IC95%)	*p*	^d^ ρ	*p*
**Elite Men**						
SPI	0.633 (0.661–0.653)	<0.001	0.633 (0.611–0.653)	<0.001	0.414	<0.001
T1PI	0.094 (0.049–0.218)	<0.001	0.190 (0.142–0.236)	<0.001	0.195	<0.001
CPI	0.774 (0.689–0.829)	<0.001	0.799 (0.787–0.810)	<0.001	0.537	<0.001
T2PI	0.084 (0.070–0.224)	<0.001	0.231 (0.184–0.275)	<0.001	0.270	<0.001
RPI	0.536 (0.184–0.788)	<0.001	0.755 (0.741–0.769)	<0.001	0.842	<0.001
**Elite Women**						
SPI	0.615 (0.587–0.640)	<0.001	0.618 (0.593–0.642)	<0.001	0.425	<0.001
T1PI	0.184 (0.052–0.361)	<0.001	0.305 (0.259–0.349)	<0.001	0.275	<0.001
CPI	0.815 (0.635–0.890)	<0.001	0.856 (0.847–0.865)	<0.001	0.705	<0.001
T2PI	0.106 (0.081–0.268)	<0.001	0.265 (0.216–0.312)	<0.001	0.289	<0.001
RPI	0.582 (0.188–0.821)	<0.001	0.794 (0.780–0.807)	<0.001	0.790	<0.001

SPI, Swimming Performance Indicator; T1PI, Transition 1 Performance Indicator; CPI, Cycling Performance Indicator; T2PI, Transition 2 Performance Indicator; RPI, Running Performance Indicator; OPI, Overall Performance Indicator; ^a^ Absolute Agreement Intraclass Correlation Coefficient; ^b^ 95% Confidence Intervals; ^c^ Concordance Intraclass Correlation Coefficient; ^d^ Spearman Range Correlation Coefficient.

**Table 4 biology-11-00902-t004:** Intraclass correlation coefficient of absolute agreement and concordance, and Spearman range correlation coefficient, between the overall performance indicator and the three segments in the years 2012–2019.

	ICC_a_ (IC95%) ^b^	*p*	ICC_c_ (IC95%)	*p*	^d^ ρ	*p*
**Elite Men**						
SPI	0.640 (0.622–0.658)	<0.001	0.640 (0.622–0.658)	<0.001	0.424	<0.001
T1PI	0.090 (0.047–0.210)	<0.001	0.184 (0.142–0.223)	<0.001	0.188	<0.001
CPI	0.777 (0.700–0.828)	<0.001	0.800 (0.789–0.809)	<0.001	0.535	<0.001
T2PI	0.083 (0.070–0.222	<0.001	0.229 (0.189–0.268)	<0.001	0.266	<0.001
RPI	0.538 (0.179–0.788)	<0.001	0.754 (0.741–0.766)	<0.001	0.843	<0.001
**Elite Women**						
SPI	0.613 (0.588–0.637)	<0.001	0.618 (0.596–0.638)	<0.001	0.425	<0.001
T1PI	0.178 (0.050–0.352)	<0.001	0.298 (0.257–0.336)	<0.001	0.277	<0.001
CPI	0.816 (0.633–0.890)	<0.001	0.857 (0.849–0.865)	<0.001	0.709	<0.001
T2PI	0.107 (0.082–0.271)	<0.001	0.269 (0.226–0.309)	<0.001	0.294	<0.001
RPI	0.581 (0.188–0.820)	<0.001	0.793 (0.781–0.804)	<0.001	0.789	<0.001

SPI, Swimming Performance Indicator; T1PI, Transition 1 Performance Indicator; CPI, Cycling Performance Indicator; T2PI, Transition 2 Performance Indicator; RPI, Running Performance Indicator; OPI, Overall Performance Indicator; ^b^ 95% Confidence Intervals; ^d^ Spearman Range Correlation Coefficient.

## Data Availability

All result data used in the analysis in this paper are available from the World Triathlon website: https://www.triathlon.org (accessed on 19 March 2020).

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
