# Peer review of "Concordance Analysis between the Segments and the Overall Performance in Olympic Triathlon in Elite Triathletes"

_biology, 2022, doi:10.3390/biology11060902_

Round 1
Reviewer 1 Report
This paper aims to determine the performance in triathlon based on the performance of its specific disciplines. Authors use a great number of entries from results in elite competitions, so I congratulate the authors for this. Nevertheless, I have also some concerns that I will mention in the next lines. My main concern is about the novelty of this study and the application of these results in relation with previous researches.
Furthermore, in the next lines I will make some comments that I hope they would be useful for the paper quality improvement.
Page 1, line 2: I think title is good and it explains clearly the paper. If you prefer a shorter option, here you have a couple of suggestions: “Concordance Analysis between the Segments and the Overall Performance in Olympic Triathlon in Elite Athletes” and “Olympic Triathlon Segments and the Overall Performance Concordance Analysis in Elite Athletes”
Page 2, line 16: The first two statements seem to be too ambitious. The performance in research in triathlon has been measured by several means (i.e., lab tests). Position is what defines de medals so it is very important. Also, time is what defines position and the aim of an athlete (swim, cycle and run faster). So, I suggest starting the paper with two less ambitious and more realistic sentences.
Page 2, line 30: I am not sure which is the predicate of this sentence. Please re-write it: “In total, 11263 entries, 6273 of 29 which were elite men and 4990 elite women.”
Page 2, line 37: Last sentence of the abstract is also over-ambitious. The performance in triathlon is composed by the three segments and thus this is the main aim of the athlete, trainer,...
Page 2, from line 42: I believe introduction could be shortened in order to highlight what is known about this issue and what is the gap that this research will cover.
Page 2, lines 43-70: The first and second paragraphs could be shortened or even deleted in some parts.
Page 2, lines 71-95: The third paragraph is very important and it resumes some important papers about this issue. Anyway, I miss a sentence at the end of the paragraph explaining what is not known about this issue (the gap).
Page 3, lines 113-136: The difference between a concordance study and an agreement / correlation study should be described more precisely and clearly.
Page 3, lines 135-144: I believe that the four hypotheses could be replaced by one paper’s objective.
Page 3, until line 145: In general, I think the introduction should be shorter, clearer and more concrete.
Page 4, line 152: In the sentence “Were examined 11263 entries”. I think a sentence kind “subject plus verb plus predicate” would be better to be understood.
Page 4, point 2.1: I am aware that some important information about athletes characteristics (mean age, training hours per week,…) is not possible to obtain. Anyway, I miss a description of participants (athletes) as much as possible: minimum age to compete in WTS,… Also, if you are able to calculate the number of athletes (not only entries), it could be a valuable information.
Page 7, until line 253: In the abstract you mentioned “this methodological study” but I am not sure if this is a methodological study since you do not compare the results of this kind of analyses (OPI, SPI, CPI,…) with other measures from previous researches that you suggest to be improved (i.e.: measurement by position) in terms of reliability and validity. Please explain the “methodological study” means in methods or introduction.
Page 12, line 396: Did you perform a separate analysis for Olympic and Sprint distance? If you did not so, maybe you cannot stat “these results support the external validation of the performance indicator at Olympic distance but also at Sprint distance”.
Page 13, line 450: I think that reference number 9 has some missing information (journal).
Page 13, line 459: I realized that some references include DOI but others do not. For example, I found that paper referenced in 14th position actually has a DOI that is not included here: DOI: 10.1123/ijspp.6.4.443
Page 13, line 474: I think that reference number 20 has some missing information (journal).
I congratulate the authors for their effort in this research. I hope my comments are helpful.
Reviewer 2 Report
Dear Authors,
Thank you for the opportunity to review the article “An in-depth Assessment of Performance in Elite Triathletes: Concordance Analysis between the Performance of the Olympic Triathlon Segments and Overall Performance”. The theme of the article is interesting for sport scientists and for coaches. The purpose of the study is to assess the degree of concordance of performance between each of the triathlon disciplines with overall performance in elite triathletes in Olympic distance triathlon. My general thoughts on the study: The introduction is described in a logical way and helps to understand the hypotheses, however it seems long to me at times and approaches a literature review, my suggestion is to reduce mainly paragraphs 3 and 6 and review the text as a whole to facilitate the reading. Still within this chapter, I believe that performance has been discussed, but the authors could define the term according to the specific literature and adopt a definition for what they consider performance to be.The methods are clear and based on an article already published by the authors about sprint triathlon. The results are clear and the figures are easy to read, I just suggest improving the image quality. The discussion follows the logic of the results and is well presented, by reading the introduction I expected to find a little more in-depth discussion of the differences between the performance of men and women, mainly linked to lines 96-103.
Specific details:
- Line 54: “From a theoretical point of view, performance is a concept that refers to the relationship between the means used to achieve something and the result that is finally obtained.”
- Line 113- 124: I believe that the comparison between correlational and concordance statistics should be the focus of the discussion. My suggestion is to shorten the explanation on Pearson's linear correlation.
- Hypotheses 2 and 3 seem contradictory, if the authors have the hypothesis that one modality has the lowest agreement with the general performance, could the other modality not have the lowest agreement?
- Line 343-345: “It means that a good cycling performance would explain a good overall performance. Therefore, the cycling performance indicator in the cycling segment could be key to anticipating overall competition performance…” In practical point of view, do you think this is possible?
- Suggestion, include in chapter 6 a practical application of the index used.
Round 2
Reviewer 1 Report
I would like to thanks the authors their comments and their effort to improve the paper following my suggestions. I let the editor know that I am not an expert in triathlon so I cannot know in what extend this paper is relevant for this sport. So, I focused my review in other issues and I think the paper is quite good now. I think authors should double-check again the paper to see some minor issues as the verb tenses, as sometimes in the same paragraph you use both the past and the present to refer to this study.
Good job!